# Multimodal Medical Code Tokenizer

**Xiaorui Su** [1]  **Shvat Messica** [1]  **Yepeng Huang** [1]  **Ruth Johnson** [1]  **Lukas Fesser** [1]  **Shanghua Gao** [1]
**Faryad Sahneh** [2]  **Marinka Zitnik** [1]

## Abstract

Foundation models trained on patient electronic health records (EHRs) require tokenizing medical data into sequences of discrete vocabulary items. Existing tokenizers treat medical codes from EHRs as isolated textual tokens. However, each medical code is defined by its textual description, its position in ontological hierarchies, and its relationships to other codes, such as disease co-occurrences and drug-treatment associations. Medical vocabularies contain more than 600,000 codes with critical information for clinical reasoning. We introduce MEDTOK, a multimodal medical code tokenizer that uses the text descriptions and relational context of codes. MEDTOK processes text using a language model encoder and encodes the relational structure with a graph encoder. It then quantizes both modalities into a unified token space, preserving modality-specific and cross-modality information. We integrate MEDTOK into five EHR models and evaluate it on operational and clinical tasks across in-patient and out-patient datasets, including outcome prediction, diagnosis classification, drug recommendation, and risk stratification. Swapping standard EHR tokenizers with MEDTOK improves AUPRC across all EHR models, by 4.10% on MIMIC-III, 4.78% on MIMIC-IV, and 11.32% on EHRShot, with the largest gains in drug recommendation. Beyond EHR modeling, we demonstrate using MEDTOK tokenizer with medical QA systems. Our results demonstrate the potential of MEDTOK as a unified tokenizer for medical codes, improving tokenization for medical foundation models.

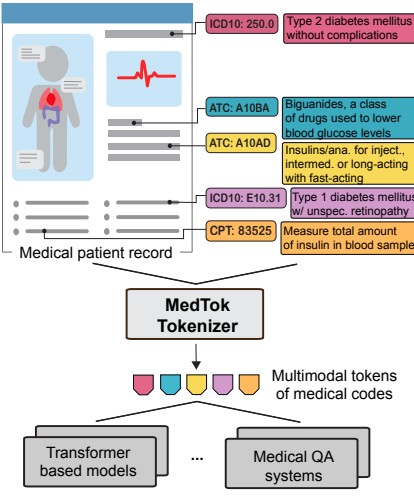

*Figure 1.* MEDTOK is a multimodal tokenizer of medical codes that combines text descriptions of codes with graph-based representations of dependencies between codes derived from clinical ontologies and standard medical terminologies. MEDTOK is a general-purpose tokenizer that can be integrated into any transformer-based model or system that requires tokenization.

## 1. Introduction

Electronic health records (EHRs) are the backbone of modern healthcare, capturing a person's health state with increasing precision across diverse modalities. Structured EHR data, encoded through standardized medical codes, support a wide range of applications, from personalized risk prediction (Goldstein et al., 2016; Yu et al., 2024b) and disease trajectory modeling (Jensen et al., 2017; Heumos et al., 2024) to emulation of clinical trials (Katsoulakis et al., 2024; Kraljevic et al., 2024). The cornerstone of structured EHRs is medical coding systems, which assign standardized alphanumeric codes to various aspects of patient health, including diseases, procedures, medications, and laboratory tests. These codes come from widely used terminologies such as ICD-9, ICD-10, SNOMED CT, CPT, and ATC, among others (Foley et al., 1992; Organization et al., 1988; Organization, 2004; Donnelly et al., 2006; Dotson, 2013; Miller & Britt, 1995). Although essential for interoperability, medical codes introduce challenges for models, particularly in the tokenization process, which transforms structured EHR data into token sequences that foundation

[1]Department of Biomedical Informatics, Harvard University, Boston, MA, USA [2]Digital Data, Sanofi, Cambridge, MA, USA. Correspondence to: Marinka Zitnik <marinka@hms.harvard.edu>.

*Proceedings of the 42nd International Conference on Machine Learning*, Vancouver, Canada. PMLR 267, 2025. Copyright 2025 by the author(s).

models can process.

Transformer-based models for structured EHRs (Poulain & Beheshti, 2024; Yang et al., 2023b; Jiang et al., 2023b; Renc et al., 2024) rely on tokenizers to map raw data into discrete vocabulary items. However, standard tokenization strategies inherited from general-purpose language models fail to capture the complexity of medical codes, leading to six key challenges: (1) Scalability of medical vocabularies – Medical coding systems contain over 600,000 unique codes, far exceeding standard tokenizer capacities. Treating each code as a separate token leads to inefficient vocabulary expansion, increasing memory demands and fragmenting rare codes (e.g., splitting "ICD9: 250.0" into arbitrary subwords). (2) Loss of hierarchical and relational structure – Many coding systems encode structured dependencies, such as ATC codes, which classify drugs based on pharmacological and chemical properties (Miller & Britt, 1995). Standard tokenizers, relying only on co-occurrence statistics, fail to capture hierarchical relationships, losing dependencies like disease co-occurrences and drug contraindications. (3) Redundancy across coding systems – Identical clinical concepts often appear under different codes across terminologies (e.g., ICD vs. SNOMED). Standard tokenization treats them as separate tokens, creating redundancy and complicating cross-system data integration. (4) Inefficiency in token storage – Expanding vocabulary sizes to accommodate medical codes results in bloated embedding tables that degrade computational efficiency, particularly for low-resource codes that appear infrequently but still require dedicated tokens. (5) Sparse and inconsistent usage – Many medical codes are rarely used or inconsistently documented, making it difficult for standard tokenizers to learn meaningful representations. Low-frequency codes suffer from poor embeddings, reducing performance on underrepresented conditions. (6) Lack of multimodal representations – Existing methods (Jiang et al., 2023b; Zhu et al., 2024; Xu et al., 2024) treat medical codes as isolated textual tokens, discarding graph-based relationships that encode essential links between diagnoses, treatments, and medications. A robust tokenizer must integrate both textual and relational information to fully represent medical codes.

Several models attempt to enrich the representations of medical codes by incorporating external knowledge from Large Language Models (LLMs) (Jiang et al., 2023b; Zhu et al., 2024; Xu et al., 2024). Methods like GraphCare and RAM-EHR prompt LLMs to generate structured knowledge triplets of medical codes or summarize retrieved knowledge. Although effective in specific tasks, these approaches suffer from limited generalizability, a heavy reliance on knowledge generated by LLMs, and a lack of a unified framework for handling various medical coding systems. Despite advances in medical representation learning, a unified tokenizer that integrates textual and structured relational knowledge across

coding systems remains an open challenge.

**Present work.** We introduce MEDTOK (https://github.com/mims-harvard/MedTok) , a multimodal medical code tokenizer that integrates textual descriptions and graph-based dependencies from biomedical ontologies (Figure 1). Unlike standard tokenization methods that treat medical codes as isolated textual tokens, MEDTOK captures both semantic meaning and structured relationships by encoding multiple modalities into a unified token space. MEDTOK operates in three stages. Multimodal encoding first extracts text embeddings from medical code descriptions and graph-based representations from biomedical knowledge graphs using separate encoders. Next, vector quantization maps both modalities into a shared token space, generating distinct text-informed and graph-informed token embeddings while preserving cross-modality interactions. Finally, optimization for expressivity ensures that token representations capture hierarchical relationships, semantic equivalence across different coding systems, and dependencies such as comorbidities and drug interactions.

We integrate MEDTOK into five EHR models and evaluate it in clinical and operational tasks that span the inpatient (MIMIC-III, MIMIC-IV) and outpatient (EHRShot) settings. These tasks include disease prediction, operational outcome modeling, drug recommendation, patient risk stratification, and operational outcomes. Our key contributions are:

- Multimodal tokenization of medical codes – MEDTOK tokenizer jointly encodes both textual descriptions and graph-based representations of medical codes, enabling richer and structured embeddings.
- Improved cross-system generalization – By incorporating ontological knowledge, MEDTOK bridges semantic gaps between different coding systems.
- Demonstrated performance gains – Replacing standard EHR tokenizers with MEDTOK improves AUPRC by 4.10% on MIMIC-III, 4.78% on MIMIC-IV, and 11.30% on EHRShot, with the largest gains in drug recommendation tasks. MEDTOK is a general purpose tokenizer that can be integrated into any transformer-based model or system that requires tokenization. Beyond EHR models, we demonstrate its applicability in medical question-answering systems, further highlighting the benefit of optimized tokenization of structured medical data.

## 2. Related work

**Domain-specific tokenizers.** Tokenizers tailored for specific domains have been employed to process various types of data, including language (Sennrich et al., 2016; Kudo & Richardson, 2018; Song et al., 2021; Wang et al., 2024b; Minixhofer et al., 2024), images (Zhou et al., 2022; Yu et al., 2022; 2024a; Zha et al., 2024), videos (Choudhury et al.,

2024), graphs (Perozzi et al., 2024; Yang et al., 2024a), and molecular and material sciences (Fu et al., 2024; Tahmid et al., 2024; Qiao et al., 2024). While these tokenizers perform well within their respective domains, they are not directly applicable to medical codes, which contains specialized medical semantics. Medical codes reside in relation contexts and are accompanied by textual descriptions. Directly using the tokenizers for languages risks flattening the relationships among codes and failing to preserve the biomedical information. This will lead to fragmented tokenization of medical codes, resulting in loss of contextual information during encoding. Meanwhile, visual tokenizer typically focus on local pixel-level relationships, which are insufficient for capturing the complex semantics inherent in medical codes. Graph tokenizers are designed to encode structured information from graphs into a discrete token, then enabling LLMs to process relational and topological knowledge effectively. However, graph tokenizers may suffer from information loss when applied to graphs in other domains, making them less flexible and efficient for large, dynamic, and cross-domain graphs. In contrast, our MED-TOK tokenizer explicitly incorporates the relevant medical semantics by integrating textual descriptions with graph-based relational contexts.

**Vector-quantized tokenizers.** Tokenization strategies often vary according to the problem domain and data modality where recent work has highlighted the benefits of discrete tokenization (Du et al., 2024). This process involves partitioning the input according to a finite set of tokens, often held in a *codebook* (this concept is independent of medical coding despite the similar name), and the quantization process involves learning a mapping from input data to the optimal set of tokens according to a pre-defined objective such as reconstruction loss (Van Den Oord et al., 2017). Recent work has highlighted the ability of vector quantized (VQ-based) tokenization to effectively compress semantic information (Gu et al., 2024). This approach is particularly successful for tokenizing inputs with an inherent semantic structure such as graphs (Yang et al., 2023a; Wang et al., 2024c), speech (Zeghidour et al., 2021; Baevski et al., 2019), and time (Yu et al., 2021) as well as complex tasks like recommendation retrieval (Wang et al., 2024d; Rajput et al., 2023; Sun et al., 2024) and image synthesis (Zhang et al., 2023; Yu et al., 2021). Another significant advantage to VQ-based tokenization is the natural integration of multiple modalities. By learning a shared latent space across modalities, each modality can jointly modeled using a common token vocabulary (Agarwal et al., 2025; Yu et al., 2023). TokenFlow leverages a dual-codebook design that allows for correlations across modalities through a dual encoder (Qu et al., 2024).

**Structured EHR, transformer-based, and foundation models.** Structured EHR models leverage patient records to learn representations for clinical prediction and operational healthcare tasks. These models differ from medical LLMs (Singhal et al., 2025; Tu et al., 2024; Singhal et al., 2023), which are typically trained on free-text clinical notes (Jiang et al., 2023a) and biomedical literature rather than structured EHR data. BEHRT (Li et al., 2020) applies deep bidirectional learning to predict future medical events, encoding disease codes, age, and visit sequences using self-attention. TransformEHR (Yang et al., 2023b) adopts an encoder-decoder transformer with visit-level masking to pretrain on EHRs, enabling multi-task prediction. GT-BEHRT (Poulain & Beheshti, 2024) models intra-visit dependencies as a graph, using a graph transformer to learn visit representations before processing patient-level sequences with a transformer encoder. Other models enhance EHR representations with external knowledge. GraphCare (Jiang et al., 2023b) integrates LLMs and biomedical knowledge graphs to construct patient-specific graphs processed via a Bi-attention Augmented Graph Neural Network. MulT-EHR (Chan et al., 2024) introduces multi-task heterogeneous graph learning with causal denoising to address data heterogeneity and confounding effects. ETHOS (Renc et al., 2024) tokenizes patient health timelines for transformer-based pretraining, achieving zero-shot performance. While these models focus on learning patient representations, MEDTOK serves a different role as a medical code tokenizer. It can be integrated into any structured EHR, transformer-based, or other foundation model, improving how medical codes are tokenized before being processed. Unlike these models, which rely on predefined tokenization schemes, MEDTOK optimizes the tokenization process itself.

## 3. Approach

MEDTOK is a multimodal medical tokenizer that leverages both text descriptions and relational contexts of medical codes. MEDTOK operates as a *tokenization function* $f(\cdot)$ that maps a medical code $m \in \mathcal{M}$ to a sequence of elements $\mathcal{T}$ in the *vocabulary* $\mathcal{V}$ with a size of $N$ by leveraging both its textual definition $\mathcal{D}(m)$ and a subgraph $\mathcal{G}(m)$ extracted from a biomedical knowledge graph $G$. Here, $\mathcal{M}$ is a set of 617,490 medical codes from eight medical coding systems: ICD-9, ICD-10-CM, ICD-10-PCS, SNOMED CT, ATC, NDC, CPT, and RxNORM.

**Problem definition.** Our goal is to train a multimodal tokenizer $f(\cdot)$ so that $\mathcal{T} = f(\mathcal{D}(m), \mathcal{G}(m))$, where $\mathcal{T} = [t_1, t_2, ..., t_T]$ and $t_i \in \mathcal{V}, 1 \le i \le T$. Then the generated $\mathcal{T}$ for medical code $m$ could be integrated to any EHR-based models $h(\cdot)$ and LMs or LLMs $p(\cdot)$ to perform predictive or generative tasks.

Figure 2 illustrates the architecture of MEDTOK, which takes both the medical code description and contextual

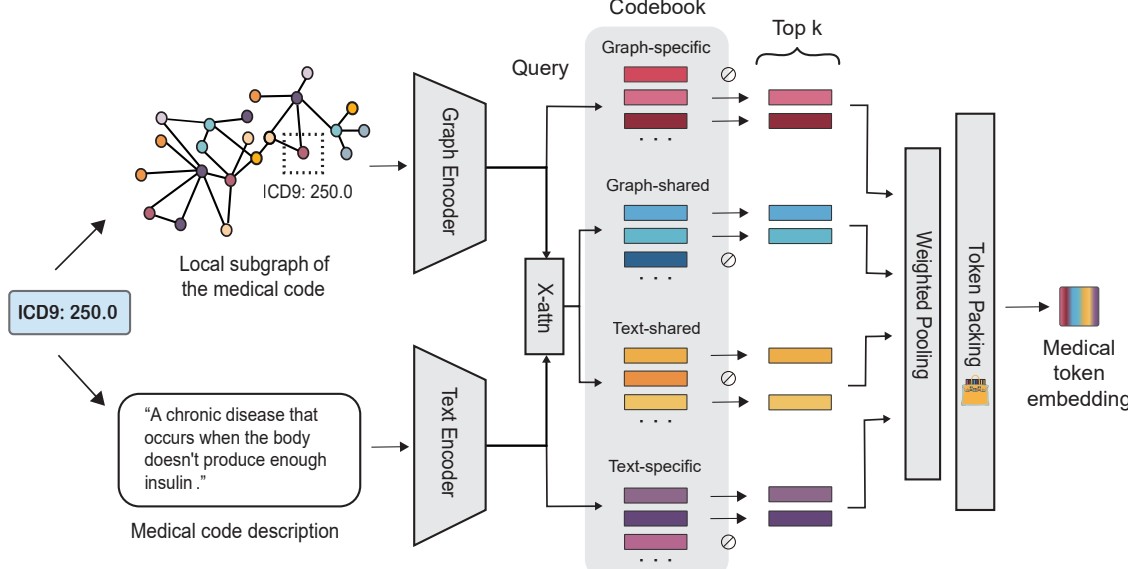

*Figure 2.* MEDTOK is a general multimodal tokenizer of medical codes that can be integrated into any transformer-based model or a system that requires tokenization. 'X-attn' denotes a cross-attention module.

knowledge from biomedical KGs as input. MEDTOK takes two steps, multimodal tokenization and token packing.

### 3.1. Multimodal tokenization

Given a medical code $m$, paired with its description $t$ and its biological subgraph $G$, MEDTOK first adopts the text encoder, denoted as $\mathrm{E}_t$ and the graph encoder, denoted as $\mathrm{E}_g$, to generate two embeddings: the text semantic embedding $\mathbf{x}_t \in \mathbb{R}^{d_t}$ for $t$ and the graph-level embedding $\mathbf{x}_g \in \mathbb{R}^{d_g}$ for $G$. These embeddings are computed as $\mathbf{x}_t = \mathrm{E}_t(t)$ and $\mathbf{x}_g = \mathrm{E}_g(G)$, where $\mathrm{E}_t$ and $\mathrm{E}_g$ represent the text and graph encoders, respectively.

**Modality-specific embeddings.** MEDTOK then adopts two linear projectors: $f_t : \mathbb{R}^{d_t} \rightarrow \mathbb{R}^d$ and $f_g : \mathbb{R}^{d_g} \rightarrow \mathbb{R}^d$, to generate modality-specific embeddings $\mathbf{e}_t^s \in \mathbb{R}^d$ and $\mathbf{e}_g^s \in \mathbb{R}^d$, respectively, where $\mathbf{e}_t^s = f_t(\mathbf{x}_t)$, $\mathbf{e}_g^s = f_g(\mathbf{x}_g)$, and $d$ is the dimension of specific embeddings.

**Cross-modality embeddings.** Moreover, MEDTOK incorporates a cross-attention module to derive cross-modality embeddings $\mathbf{e}_t^c \in \mathbb{R}^d$ and $\mathbf{e}_g^c \in \mathbb{R}^d$, Specifically, the embedding $\mathbf{e}_t^c$ is computed as:

$$\mathbf{e}_t^c = \mathrm{softmax}\left( \frac{\mathbf{W}_q^t \mathbf{x}_t (\mathbf{W}_k^g \mathbf{x}_g)^T}{\sqrt{d}} \right) (\mathbf{W}_v^g \mathbf{x}_g) \quad (1)$$

where $\mathbf{W}_q^t \in \mathbb{R}^{d \times d_t}, \mathbf{W}_k^g \in \mathbb{R}^{d \times d_g}$, and $\mathbf{W}_v^g \in \mathbb{R}^{d \times d_g}$ represent the query, key and value weight matrix. Similarly, the embedding $\mathbf{e}_g^c$ is given by:

$$\mathbf{e}_g^c = \mathrm{softmax}\left( \frac{\mathbf{W}_q^g \mathbf{x}_g (\mathbf{W}_k^t \mathbf{x}_t)^T}{\sqrt{d}} \right) (\mathbf{W}_v^t \mathbf{x}_t) \quad (2)$$

where $\mathbf{W}_q^g \in \mathbb{R}^{d \times d_g}, \mathbf{W}_k^t \in \mathbb{R}^{d \times d_t}$, and $\mathbf{W}_v^t \in \mathbb{R}^{d \times d_t}$ represents the query, key, and value weight matrix.

**Tokenization.** After generating modality-specific and cross-modality embeddings, for each embedding, MEDTOK quantizes the embedding into $K$ tokens by querying a unified *codebook* $\mathbf{C} \in \mathbb{R}^{N \times d}$. The $K$ tokens are identified by the top $K$ nearest vectors in the *codebook*.

In detail, for any modality-specific or cross-modality embedding $\mathbf{e}_:$, its quantized tokens $\mathcal{I}(\mathbf{e}_:)$ is formulated by:

$$\mathcal{I}(\mathbf{e}_:) = \mathrm{argmin}_K \left( \{\mathrm{dist}(\mathbf{e}_:, \mathbf{C}_i)\}_{i=1}^N \right) \quad (3)$$

where $\mathrm{dist}(:, :)$ denotes the Euclidean distance, $|\mathcal{I}(\mathbf{e}_:)| = K$, and $\mathbf{C}_i = \mathbf{C}[i, :]$. Then MEDTOK assigns a weight to each token $k \in \mathcal{I}(\mathbf{e}_:)$ based on the distance between $\mathbf{e}_:$ and its corresponding vector $\mathbf{C}_k = \mathbf{C}[k, :]$. These weighted tokens are then summed together to obtain the quantized vector for $\mathbf{e}_:$, denoted as $\hat{\mathbf{e}}_:$, which is given by:

$$\hat{\mathbf{e}}_: = \sum_{k \in \mathcal{I}(\mathbf{e}_:)} -\mathrm{softmax}(\mathrm{dist}(\mathbf{e}_:, \mathbf{C}_k)) * \mathbf{C}_k \quad (4)$$

Following vector quantization conventions, we employ a straight-through gradient estimator: $\mathbf{e}_: = \mathrm{sg}[\mathbf{e}_: - \hat{\mathbf{e}}_:] + \hat{\mathbf{e}}_:$ where $\mathrm{sg}[\cdot]$ denotes the stop-gradient operation. The *codebook* learning objective is $\mathcal{L}(\mathbf{e}_:, \hat{\mathbf{e}}_:) = \|\mathrm{sg}[\hat{\mathbf{e}}_:] - \mathbf{e}_:\|_2^2 + \alpha * \|\hat{\mathbf{e}}_: - \mathrm{sg}[\mathbf{e}_:]\|_2^2$, where $\alpha$ is a hyperparameter.

To preserve the distinctiveness of modality-specific and cross-modality embeddings, MEDTOK divides the entire *codebook* into three regions: a text-specific region, a graph-specific region, and a shared region. The shared region

includes the graph-shared and text-shared region and is shared with both two modalities. MEDTOK then queries distinct regions of the *codebook* to generate their respective tokens and quantized vectors, which are represented by: $(\mathcal{I}(\mathbf{e}_t^s), \mathcal{I}(\mathbf{e}_g^s), \mathcal{I}(\mathbf{e}_t^c), \mathcal{I}(\mathbf{e}_g^c))$ and $(\hat{\mathbf{e}}_t^s, \hat{\mathbf{e}}_g^s, \hat{\mathbf{e}}_t^c, \hat{\mathbf{e}}_g^c)$. The final *codebook* objective is as follows:

$$\mathcal{L}_C = \mathcal{L}(\mathbf{e}_t^s, \hat{\mathbf{e}}_t^s) + \mathcal{L}(\mathbf{e}_g^s, \hat{\mathbf{e}}_g^s) + \mathcal{L}(\mathbf{e}_t^c, \hat{\mathbf{e}}_t^c) + \mathcal{L}(\mathbf{e}_g^c, \hat{\mathbf{e}}_g^c) \quad (5)$$

### 3.2. Token packing

Unlike image-text paired data, where modalities have substantial overlap, the two modalities considered here (text and graph representations of medical codes) are more distinct but also highly complementary. Text provides the clinical definitions and describes each medical code in natural language. In contrast, the graph representation encodes domain-specific relationships such as disease co-occurrence, hierarchical groupings, and other medical ontologies. These relationships are not fully captured by text alone and introduce structured, expert-driven knowledge that is critical for many clinical and scientific applications. By considering both modalities, MEDTOK generates a representation that captures the shared information and also preserves information that is unique to each modality. MEDTOK achieves this by extracting modality-specific features during the tokenization process, rather than relying on standard approaches that may blend or discard valuable distinctions. This ensures that MEDTOK's tokens reflect both clinical language and the structured relationships present in the graph.

We pack tokens $(\mathcal{I}(\mathbf{e}_t^s), \mathcal{I}(\mathbf{e}_g^s), \mathcal{I}(\mathbf{e}_t^c), \mathcal{I}(\mathbf{e}_g^c))$ by optimizing both modality-shared and modality-specific information between these tokens and their corresponding quantized vectors. To capture modality-shared information, we focus on the tokens $\mathcal{I}(\mathbf{e}_t^c)$ and $\mathcal{I}(\mathbf{e}_g^c)$. The objective uses Kullback-Leibler divergence to align their distance matrices, $\text{dist}(\mathbf{e}_t^c, \mathbf{C})$ and $\text{dist}(\mathbf{e}_g^c, \mathbf{C})$, such that they follow a similar distribution: $\mathcal{L}_{\text{KL}} = D_{\text{KL}}(\text{softmax}(\text{-dist}(\mathbf{e}_t^c, \mathbf{C})) \parallel \text{softmax}(\text{-dist}(\mathbf{e}_g^c, \mathbf{C})))$. Next, we optimize the quantized vectors $\hat{\mathbf{e}}_t^c$ and $\hat{\mathbf{e}}_g^c$ to maximize the information they carry about the other modality, while minimizing redundancy with their own modality. Specifically, we solve:

$$\hat{\mathbf{e}}_t^{c*} = \arg\max_{\hat{\mathbf{e}}_t^c} \left( I(\hat{\mathbf{e}}_t^c; \mathbf{e}_g^c) - \beta \cdot I(\hat{\mathbf{e}}_t^c; \mathbf{e}_t^c | \mathbf{e}_g^c) \right) \quad (6)$$

$$\hat{\mathbf{e}}_g^{c*} = \arg\max_{\hat{\mathbf{e}}_g^c} \left( I(\hat{\mathbf{e}}_g^c; \mathbf{e}_t^c) - \beta \cdot I(\hat{\mathbf{e}}_g^c; \mathbf{e}_g^c | \mathbf{e}_t^c) \right) \quad (7)$$

For modality-specific information, MEDTOK optimizes tokens $\mathcal{I}(\mathbf{e}_t^s)$ and $\mathcal{I}(\mathbf{e}_g^s)$ by ensuring that the quantized vectors $\hat{\mathbf{e}}_t^s$ and $\hat{\mathbf{e}}_g^s$ retain maximal information about their respective modalities while minimizing shared information between modalities. The optimal solutions are given by:

$$\hat{\mathbf{e}}_t^{s*} = \arg\max_{\hat{\mathbf{e}}_t^s} \left( I(\hat{\mathbf{e}}_t^s, \hat{\mathbf{e}}_g^c; \mathbf{e}_t^s) - \lambda \cdot I(\mathbf{e}_t^s; \hat{\mathbf{e}}_g^{c*}) \right) \quad (8)$$

$$\hat{\mathbf{e}}_g^{s*} = \arg\max_{\hat{\mathbf{e}}_g^s} \left( I(\hat{\mathbf{e}}_g^s, \hat{\mathbf{e}}_t^c; \mathbf{e}_g^s) - \lambda \cdot I(\mathbf{e}_g^s; \hat{\mathbf{e}}_g^{c*}) \right) \quad (9)$$

Based on the derivation of Wang et al., the loss for packing shared information across two modalities is formulated by: $\mathcal{L}_{token}^c = \mathcal{L}_{\text{InfoNCE}}(\hat{\mathbf{e}}_t^c, \hat{\mathbf{e}}_g^c) + \mathcal{L}_{\text{InfoNCE}}(\hat{\mathbf{e}}_g^c, \hat{\mathbf{e}}_t^c) - 2\beta\mathbb{E}_{\mathbf{e}_t^c, \mathbf{e}_g^c}(\mathbf{e}_t^c \cdot \mathbf{e}_g^c)$, where $\mathcal{L}_{\text{InfoNCE}}$ denotes the InfoNCE loss. Additionally, the loss for packing specific information across two modalities is formulated by: $\mathcal{L}_{token}^s = \mathcal{L}_{\text{InfoNCE}}(\hat{\mathbf{e}}_t^c, \tilde{\mathbf{e}}_t^c) + \lambda\mathcal{L}_{\text{orthogonal}}(\hat{\mathbf{e}}_t^c, \mathbf{e}_t^c) + \mathcal{L}_{\text{InfoNCE}}(\hat{\mathbf{e}}_g^c, \tilde{\mathbf{e}}_g^c) + \lambda\mathcal{L}_{\text{orthogonal}}(\hat{\mathbf{e}}_g^c, \mathbf{e}_g^c)$, where $\mathcal{L}_{\text{orthogonal}}$ denotes the orthogonal loss.

We combine modality-shared and modality-specific losses into an overall token packing loss as: $\mathcal{L}_{token} = \mathcal{L}_{token}^c + \mathcal{L}_{token}^s$, where $\beta$ and $\lambda$ are hyperparameters set to be equal. This approach allows MEDTOK to leverage both modality-shared and modality-specific information.

### 3.3. Training and inference for MEDTOK

During the training stage, MEDTOK is trained by the sum of *codebook* loss $\mathcal{L}_C$, KL divergency loss $\mathcal{L}_{KL}$, token packing loss $\mathcal{L}_{token}$, where $\mathcal{L} = \mathcal{L}_C + \mathcal{L}_{KL} + \mathcal{L}_{token}$. After pre-training, MEDTOK can be integrated into any model or pipeline dealing with medical codes, providing unified medical tokens for downstream tasks.

## 4. Experiments

**Medical coding systems.** We collected a total of 617,490 medical codes from eight commonly used coding systems: ICD-9 (Organization et al., 1988), ICD-10-CM (Fung et al., 2020), ICD-10-PCS (Averill et al., 2001), SNOMED CT (Donnelly et al., 2006), ATC (Miller & Britt, 1995), NDC (Palmer, 2006), CPT (Dotson, 2013), and RxNORM (Nelson et al., 2011), as shown in Table 1. These codes cover various events, including procedures, diagnoses, and medications. Each code is paired with a textual description from official documents and a subgraph from PrimeKG (Chandak et al., 2023). Details are available in Appendix A.

| Code system | Count | Code system | Count |
|---|---|---|---|
| SNOMED | 303,325 | ICD9 | 18,365 |
| ICD10-CM | 81,184 | CPT | 10,602 |
| RxNorm | 81,151 | ATC | 6,659 |
| ICD10-PCS | 61,644 | NDC | 54,560 |

*Table 1.* Summary of the dataset's code systems distribution.

**Patient EHR datasets.** We used three publicly available EHR datasets: MIMIC-III (Johnson et al., 2016), MIMIC-IV (Johnson et al., 2024), and EHRShot (Wornow et al., 2023). MIMIC-III and MIMIC-IV are in-patient datasets with medical records for ICU patients, while EHRShot is a dataset containing longitudinal medical records that include both out-patients and ICU/ED patients. MIMIC datasets

include NDC medications and ICD-9 / ICD-10 codes for diagnoses and procedures. In contrast, EHRShot mainly uses RxNorm codes for medications, SNOMED codes for diagnoses, and CPT, SNOMED, ICD-9, and ICD-10 codes for procedures. Table 2 summarizes the statistics of three EHR datasets.

|  | #patients | #visits | #visits/patient | #events/patient |
|---|---|---|---|---|
| MIMIC-III | 35,707 | 44,399 | 1.24 | 51.14 |
| MIMIC-IV | 123,488 | 232,263 | 1.88 | 70.33 |
| EHRShot | 6,739 | 921,499 | 136.74 | 6182.17 |

*Table 2.* Statistics of EHR datasets.

**Baselines.** We consider two tokenizers and five EHR-based models based on EHR. The first type of tokenizer is text-based (e.g., bert-base-uncased (Devlin et al., 2018)), while the second is graph-based (e.g., VQGraph (Yang et al., 2024b)). Five EHR-based models are ETHOS (Renc et al., 2024), GT-BEHRT (Poulain & Beheshti, 2024), MulT-EHR (Chan et al., 2024), TransformEHR (Yang et al., 2023b), and BEHRT (Li et al., 2020). Details on implementation can be found in the Appendix B.

**Evaluation setup.** We consider two evaluation setups:

• **In-patient evaluation:** This setting combines the MED-TOK tokenizer with patient prediction models, using two in-patient datasets that include individuals admitted to a hospital. The evaluation encompasses five tasks: ① mortality prediction (MT), ② readmission prediction (RA), ③ length-of-stay prediction (LOS), ④ phenotype prediction (Pheno), and ⑤ drug recommendation (DrugRec). The first three tasks focus on predicting a patient's future health status using their historical medical records. Phenotype prediction involves the identification of the phenotype of a patient's disease based on their medical history. We identified 24 phenotypes for diseases in MIMIC-III and MIMIC-IV, as follows (Harutyunyan et al., 2019). Drug recommendation aims to suggest appropriate medications for a patient, considering their historical medical records and the diseases identified during their current visit. For drug recommendation, we focus on five specific drug candidates, including Vancomycin, Levofloxacin, Heparin Sodium, Metoprolol, and Atorvastatin, rather than considering the entire range of available medications. AUPRC is adopted to evaluate the model's performance on the above classification tasks.

• **Out-patient evaluation:** We evaluate MEDTOK together with patient prediction models on a dataset of patients who are not admitted to a hospital and consider two categories of tasks: ① Operational Outcomes (OO), and ② new diagnosis assignments (ND), following (Wornow et al., 2023). The OO includes MT, RA, and prolonged LOS. The new diagnosis assignments are used to predict the first diagnosis of a disease. Details are in Appendix C.

### 4.1. MEDTOK tokenizer with in-patient EHR models

Table 3 presents the AUPRC values for each baseline and their integration with our MEDTOK for five tasks in two in-patient datasets. Compared to baselines that treat each medical code as an individual token, integrating our MED-TOK consistently improves performance across all five tasks, achieving an average improvement of 3.29% on MIMIC-III and 2.67% on MIMIC-IV. This improvement comes from more informative tokens generated by MEDTOK, which strengthen the EHR-based models. Among five tasks, MED-TOK demonstrates the most significant impact on drug recommendation tasks, highlighting the value of incorporating prior knowledge into our tokenizer.

To further assess the effectiveness of MEDTOK, we compare it against two tokenization methods: the text-based BERT tokenizer and the graph-based VQGraph tokenizer. Figure 3 presents the performance of each tokenizer when integrated with a Transformer-based EHR model (TransformEHR) across five tasks on two in-patient datasets. MEDTOK consistently outperforms BERT and VQGraph in all tasks and datasets, demonstrating the superiority of its tokenization strategy.

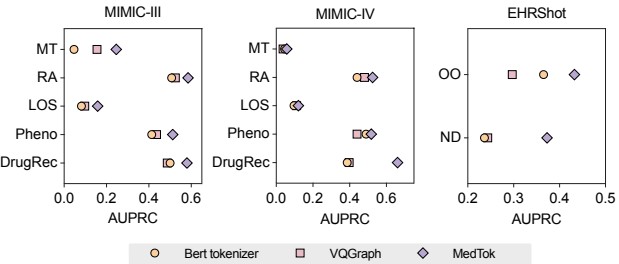

*Figure 3.* The AUPRC values of three types of tokenizers on in-patient and out-patient datasets, where OO means Operational Outcomes and ND means assignment of new diagnoses.

### 4.2. MEDTOK tokenizer with out-patient EHR models

Table 4 presents the AUPRC values for each baseline and its integration with MEDTOK across two task types on the out-patient EHRShot dataset. The results reveal that our tokenizer has the most significant impact on mortality prediction in Operational Outcomes, achieving an average improvement of 11.32%. It also significantly improves the detection of new diagnoses of Hyperlipidemia, with an average improvement of 6.00%. As shown in Figure 3, a comparison of three types of tokenizers further demonstrates the effectiveness of MEDTOK in integrating both graph and textual modalities. Additionally, when comparing performance across two in-patient datasets, we observe that MEDTOK is particularly beneficial for longitudinal data.

| Model | Task 1: MT$^+$ | | Task 2: RA($<$15 days)$^+$ | | Task 3: LOS$^*$ | | Task 4: Pheno$^\circ$ | | Task 5: DrugRec$^\circ$ | |
|---|---|---|---|---|---|---|---|---|---|---|
| | MIMIC-III AUPRC | MIMIC-IV AUPRC | MIMIC-III AUPRC | MIMIC-IV AUPRC | MIMIC-III AUPRC | MIMIC-IV AUPRC | MIMIC-III AUPRC | MIMIC-IV AUPRC | MIMIC-III AUPRC | MIMIC-IV AUPRC |
| ETHOS | 0.617 (0.010) | 0.282 (0.001) | 0.421 (0.007) | 0.648 (0.005) | N/A | N/A | N/A | N/A | 0.104 (0.008) | 0.131 (0.005) |
| + MEDTOK | **0.634 (0.020)** | **0.412 (0.030)** | **0.463 (0.017)** | **0.690 (0.007)** | N/A | N/A | N/A | N/A | **0.170 (0.014)** | **0.240 (0.012)** |
| GT-BEHRT | 0.160 (0.037) | 0.028 (0.004) | 0.612 (0.058) | 0.586 (0.070) | 0.230 (0.010) | 0.103 (0.001) | 0.423 (0.002) | 0.493 (0.005) | 0.715 (0.002) | 0.736 (0.007) |
| + MEDTOK | **0.193 (0.046)** | **0.034 (0.005)** | **0.623 (0.052)** | **0.609 (0.064)** | **0.287 (0.039)** | **0.114 (0.003)** | **0.459 (0.028)** | **0.512 (0.006)** | **0.740 (0.004)** | **0.783 (0.010)** |
| MulT-EHR | 0.136 (0.021) | 0.120 (0.003) | 0.574 (0.008) | 0.515 (0.007) | 0.176 (0.018) | 0.118 (0.032) | 0.460 (0.012) | 0.498 (0.001) | 0.523 (0.008) | 0.445 (0.027) |
| + MEDTOK | **0.156 (0.025)** | **0.141 (0.013)** | **0.585 (0.016)** | **0.565 (0.002)** | **0.198 (0.011)** | **0.136 (0.030)** | **0.480 (0.002)** | **0.504 (0.001)** | **0.571 (0.006)** | **0.465 (0.003)** |
| TransformEHR | 0.207 (0.012) | 0.042 (0.012) | 0.527 (0.030) | 0.518 (0.012) | 0.132 (0.021) | 0.119 (0.001) | 0.469 (0.022) | 0.507 (0.007) | 0.533 (0.030) | 0.612 (0.046) |
| + MEDTOK | **0.246 (0.044)** | **0.058 (0.007)** | **0.568 (0.017)** | **0.525 (0.017)** | **0.159 (0.031)** | **0.121 (0.002)** | **0.513 (0.024)** | **0.518 (0.012)** | **0.580 (0.035)** | **0.661 (0.092)** |
| BEHRT | 0.163 (0.037) | 0.028 (0.003) | 0.529 (0.053) | 0.514 (0.015) | 0.232 (0.015) | 0.112 (0.003) | 0.587 (0.004) | 0.493 (0.006) | 0.539 (0.013) | 0.778 (0.014) |
| + MEDTOK | **0.220 (0.025)** | **0.032 (0.006)** | **0.574 (0.040)** | **0.515 (0.005)** | **0.251 (0.030)** | **0.137 (0.004)** | **0.603 (0.008)** | **0.504 (0.006)** | **0.558 (0.006)** | **0.792 (0.007)** |
| *Improvement (%)* | *+3.32%* | *3.54%* | *3.00%* | *2.46%* | *3.13%* | *1.40%* | *2.90%* | *1.18%* | *4.10%* | *4.78%* |

$+$: imbalanced binary classification; $*$: multi-class classification, macro-averaged; $\circ$: multi-label classification; N/A indicates that the model was not configured for this task.

*Table 3.* The results of MEDTOK with all baseline models across five tasks on two in-patient datasets.

| Model | Task 1: Operational Outcomes (OO) | | | Task 2: Assignment of New Diagnoses (ND) | | | |
|---|---|---|---|---|---|---|---|
| | Long LOS AUPRC | RA ($<$15 days) AUPRC | MT AUPRC | Hypertension AUPRC | Hyperlipidemia AUPRC | Pancreatic Cancer AUPRC | Acute MI AUPRC |
| ETHOS | NA | 0.079 (0.017) | 0.102 (0.018) | 0.166 (0.020) | 0.155 (0.031) | 0.056 (0.006) | 0.093 (0.011) |
| + MEDTOK | NA | **0.128 (0.025)** | **0.339 (0.010)** | **0.175 (0.019)** | **0.163 (0.025)** | **0.056 (0.013)** | **0.104 (0.017)** |
| GT-BEHRT | 0.714 (0.021) | 0.115 (0.012) | 0.239 (0.012) | 0.303 (0.018) | 0.239 (0.007) | 0.044 (0.008) | 0.015 (0.008) |
| + MEDTOK | **0.739 (0.025)** | **0.154 (0.013)** | **0.444 (0.015)** | **0.360 (0.012)** | **0.441 (0.005)** | **0.074 (0.010)** | **0.031 (0.015)** |
| MulT-EHR | 0.539 (0.025) | 0.125 (0.014) | 0.397 (0.016) | 0.218 (0.005) | 0.243 (0.005) | 0.022 (0.008) | 0.017 (0.003) |
| + MEDTOK | **0.571 (0.015)** | **0.188 (0.021)** | **0.444 (0.012)** | **0.226 (0.006)** | **0.254 (0.021)** | **0.037 (0.015)** | **0.028 (0.014)** |
| TransformEHR | 0.652 (0.023) | 0.197 (0.016) | 0.344 (0.030) | 0.376 (0.018) | 0.305 (0.021) | 0.053 (0.006) | 0.025 (0.006) |
| + MEDTOK | **0.675 (0.018)** | **0.243 (0.016)** | **0.379 (0.034)** | **0.413 (0.026)** | **0.333 (0.018)** | **0.082 (0.012)** | **0.052 (0.017)** |
| BEHRT | 0.582 (0.032) | 0.332 (0.022) | 0.389 (0.018) | 0.233 (0.027) | 0.251 (0.019) | 0.036 (0.008) | 0.013 (0.031) |
| + MEDTOK | **0.723 (0.028)** | **0.397 (0.036)** | **0.431 (0.017)** | **0.287 (0.018)** | **0.302 (0.015)** | **0.057 (0.012)** | **0.036 (0.015)** |
| *Improvement (%)* | *+5.52%* | *+5.24%* | *+11.32%* | *+3.30%* | *+6.00%* | *+1.90%* | *+1.76%* |

*Table 4.* The results of MEDTOK with all baseline models across two tasks on the EHRShot dataset.

## 4.3. Ablation studies

To comprehensively understand the contributions of various components in MEDTOK, we conduct ablation studies on: (1) the effect of input modalities, and (2) the effect of shared and modality-specific optimization strategies. To ensure that our analysis is not confounded by architectural differences, we integrate MEDTOK with a standardized Transformer-based backbone (e.g., TransformEHR). This setup allows us to attribute performance differences directly to modalities and optimization strategies, rather than model architecture.

**Multimodal learning in MEDTOK.** To evaluate the impact of the two modalities (text, graph) used in MEDTOK —medical code definitions and biological subgraphs derived from a biomedical knowledge graph—we assess its performance by removing the text and graph modalities separately. As shown in Figure 4, MEDTOK, when leveraging both modalities, achieves the best performance across all tasks on three datasets. By comparing the performance of MEDTOK without the graph modality and MEDTOK without the text modality, we observe that both modalities contribute significantly to EHR-based prediction tasks. The graph modality benefits drug recommendation and new disease detection tasks, while the text modality proves essential for readmission prediction on MIMIC-III and operational outcomes in EHRShot. These findings emphasize the importance of incorporating the underlying information linked to medical codes.

**Effects of modality-shared and modality-specific information on MEDTOK.** MEDTOK is built on two modalities, and we have analyzed the impact of each modality in Figure 4. The results show that the model performs best when using both modalities. Additionally, MEDTOK optimizes tokens by maximizing shared information between modalities while preserving modality-specific information. To assess the contribution of each loss component, we conducted ablation studies by retraining MEDTOK with different loss function combinations. We then apply the pretrained MEDTOK to all tasks across datasets to obtain its performances. The results (average AUPRC across all tasks) in Table 5 demonstrate that both shared and specific information optimization enhance performance, with the full optimization achieving the best results across all datasets. The vector quantization loss $\mathcal{L}_C$ is the basic loss for tokenization. By optimizing both shared and specific information across two modalities, the performances are improved by 3.9%, 5.7%, and 9.1% on three datasets, respectively. The experimental results also show that shared and specific information contribute more to out-patient datasets.

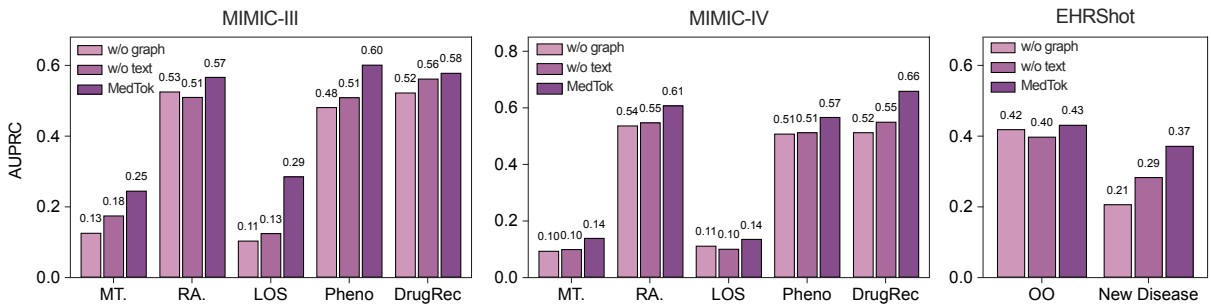

*Figure 4.* The AUPRC values obtained by removing the text and graph modalities across all tasks on two in-patient datasets and one out-patient dataset.

| Optimization | MIMIC III | MIMIC IV | EHRShot |
|---|---|---|---|
| $\mathcal{L}_C$ | 0.373 | 0.387 | 0.287 |
| $\mathcal{L}_C + \mathcal{L}_{token}^c + \mathcal{L}_{KL}$ | 0.379 | 0.409 | 0.314 |
| $\mathcal{L}_C + \mathcal{L}_{token}^s$ | 0.382 | 0.402 | 0.366 |
| $\mathcal{L}$ | 0.412 | 0.444 | 0.378 |

$\mathcal{L} = \mathcal{L}_C + \mathcal{L}_{token}^c + \mathcal{L}_{KL} + \mathcal{L}_{token}^s$

*Table 5.* The averaged AUPRC across all tasks with different loss function combinations.

### 4.4. Hyperparameter sensitivity analysis

We next investigate the impact of hyperparameters on Med-Tok's performance across all datasets. MedTok is trained with three key hyperparameters: two weighting coefficients, $\lambda$ and $\beta$, which control the contribution of shared and specific information loss components, and a *codebook* size parameter, $N$, which determines the number of discrete tokens available for representation.

**Effects of loss weight $\lambda$ and $\beta$.** To ensure that MedTok treats shared and specific information equally, we set $\lambda = \beta$, where $\lambda$ and $\beta$ are the weighting coefficients for the shared and specific information loss terms, respectively. The average AUPRC scores across all tasks are presented in below Figure 5A. The results demonstrate the influence of hyperparameter choices on model performance across the three datasets. Based on our findings, we recommend setting $\lambda = \beta = 0.1$ for in-patient settings and $\lambda = \beta = 0.01$ for out-patient settings.

**Effects of *codebook* size $N$.** We further evaluate the impact of the *codebook* size on the performance of MEDTOK by training it with varying sizes and assessing its effectiveness across three distinct datasets integrated with TransformEHR. Figure 5B presents the results for various *codebook* sizes across all tasks on the three datasets. The performance trends observed on MIMIC-III and MIMIC-IV are quite consistent, demonstrating a clear pattern where increasing the *codebook* size enhances the model's performance. Specifically, the most stable average performance is achieved when

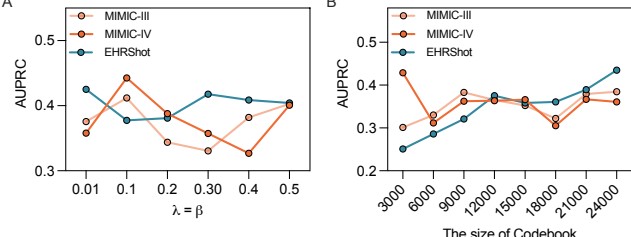

*Figure 5.* A, The AUPRC values of MEDTOK with different weighting coefficients $\lambda, \beta$; B, The AUPRC values of MEDTOK with different *codebook* size $N$.

the *codebook* size is set to $N = 12,000$, indicating that this size strikes an optimal balance between sufficient coverage of the medical vocabulary and avoiding overfitting.

In contrast, when analyzing the performance on EHRShot, a dataset consisting of patients with longer visit histories than those in MIMIC-III and MIMIC-IV, we observe that MED-TOK benefits from a larger *codebook* size. For EHRShot, the highest average performance is achieved when the *codebook* size is increased to $N = 24,000$. This suggests that for datasets with more extensive patient visit histories, a larger *codebook* may be more effective in capturing the underlying complexity of the medical information, thus improving the model's predictive capabilities.

### 4.5. Using MEDTOK tokenizer for medical QA

MEDTOK demonstrates strong performance in EHR-based tasks, as shown in Tables 3-4. To further assess its capabilities, we explore its effectiveness in a generation task, specifically multiple-choice medical question answering (MedicalQA), where the goal is to select the correct answer to a given clinical question (Singhal et al., 2023).

We evaluate whether MEDTOK enhances few-shot learning in MedicalQA by integrating its tokenized representations with three LLMs (LLaMA3.1-8B (Dubey et al., 2024), Qwen2.5-7B (Hui et al., 2024), MMedLM (Qiu et al., 2024)). MEDTOK-generated tokens are used as prefix to-

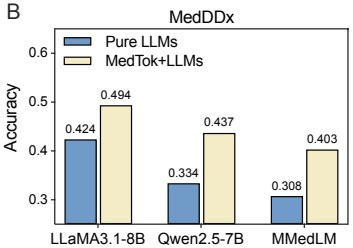

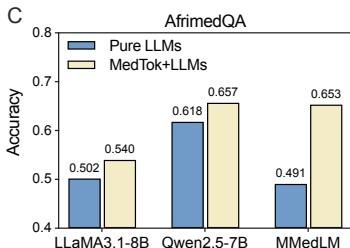

*Figure 6.* The accuracy of LLMs vs. MEDTOK+LLMs on three medical QA datasets.

kens, which provide structured medical context before the main input, allowing the LLM to incorporate medical codes.

For this evaluation, we use three medical QA datasets, including MMLU (Hendrycks et al., 2021), MedDDx (Su et al., 2024), and AfrimedQA (Olatunji et al., 2024). In addition, MedDDx dataset contains questions at three difficulty levels: Basic, Intermediate, and Expert.

The process consists of three steps: (1) Disease code mapping – Extract disease mentions from each question and retrieve their corresponding medical codes. (2) Tokenization via MEDTOK – Convert medical codes into structured tokens using MEDTOK. (3) Prefix token fine-tuning – Fine-tune LLMs using MEDTOK tokens as prefix inputs before the question text. We fine-tune the model on the MedMCQA dataset (Pal et al., 2022), and then evaluate the fine-tuned model on MMLU, MedDDx, and AfrimedQA.

The results in Figure 6 show an accuracy improvement across all datasets compared with LLMs, suggesting that MEDTOK can enhance medical QA when used as a structured representation of medical codes and integrated with an LLM through prefix tuning.

### 4.6. Interpretability of MEDTOK

To this end, we selected a subset of patients classified as high risk for Hyperlipidemia by MEDTOK+TransformEHR, where these patients have no records of Hyperlipidemia before. We then counted the tokens assigned to these patients and identified those appearing more than 100 times, as shown in Figure 7. We then mapped these frequent tokens to medical codes, with the most overlapping codes being Rosuvastatin 5 mg Oral Tablet (RxNorm 2669980), Burn of skin (SNOMED CT 147087003), Type 2 diabetes mellitus without complication (disorder) (SNOMED CT 373555004), and Hyperlipidemia (SNOMED CT 285605009). They are closely related to Hyperlipidemia. Rosuvastatin corresponds to medications commonly prescribed for lipid disorders. The other three medical codes represent clinical diagnoses or findings associated with hyperlipidemia-related cardiovascular risk. It suggests that MEDTOK captures key medical concepts related to Hyperlipidemia, supporting its predictive capability.

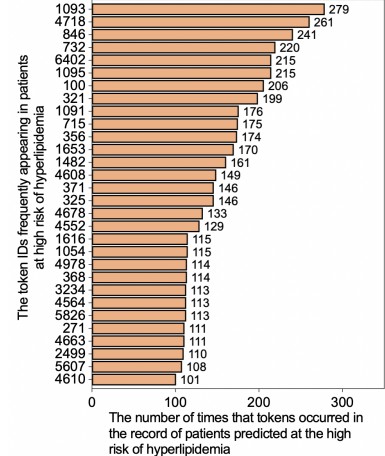

*Figure 7.* Top 100 frequent token IDs appearing in patients at high risk of Hyperlipidemia.

## 5. Conclusion

Tokenizing medical codes is a critical yet challenging step in developing foundation models for EHRs. Existing tokenizers treat medical codes as isolated textual units, failing to capture their structured relationships within large-scale medical ontologies. With more than 600,000 codes that span multiple terminologies, standard tokenization methods struggle to scale while preserving the rich semantic and relational context necessary for downstream clinical and operational tasks. We introduced MEDTOK, a multimodal tokenizer of medical codes that integrates textual definitions and relational ontologies of medical codes to create a unified token representation. MEDTOK applies vector quantization to encode both modalities in a structured token space, preserving cross-modality relationships. We integrated MEDTOK with five EHR models, evaluating its impact across inpatient (MIMIC-III, MIMIC-IV) and outpatient (EHRShot) settings, as well as in fine-tuning a medical question-answering system. Our results establish MEDTOK as a generalizable tokenizer for medical codes, shedding light on how optimizing the tokenization process can benefit medical foundation models.

## Impact statement

This work presents a tokenizer for medical codes designed to assist other models in better encoding semantic knowledge. While our tokenizer complements other models, it does not directly raise ethical concerns. Instead, it enhances the trustworthiness of these models by providing text and graph-based information as references for tokenization. In future work, we will explore ways to better integrate our tokenizer with other models to further improve both their performance and trustworthiness.

## Acknowledgement

We sincerely appreciate the valuable discussions with Intae Moon and Zhenglun Kong, whose insights and feedback have contributed to the development of this work. We gratefully acknowledge the support of NIH R01-HD108794, NSF CAREER 2339524, US DoD FA8702-15-D-0001, ARPA-H BDF program, awards from Chan Zuckerberg Initiative, Bill & Melinda Gates Foundation INV-079038, Amazon Faculty Research, Google Research Scholar Program, AstraZeneca Research, Roche Alliance with Distinguished Scientists, Sanofi iDEA-iTECH, Pfizer Research, John and Virginia Kaneb Fellowship at Harvard Medical School, Biswas Computational Biology Initiative in partnership with the Milken Institute, Harvard Medical School Dean's Innovation Fund for the Use of Artificial Intelligence, Harvard Data Science Initiative, and Kempner Institute for the Study of Natural and Artificial Intelligence at Harvard University. Any opinions, findings, conclusions or recommendations expressed in this material are those of the authors and do not necessarily reflect the views of the funders.

## Conflict of interest

Faryad Sahneh is a Sanofi employee and may hold shares and/or stock options in the company. Other authors declare no conflict of interest.

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

# A. Data preprocessing

## A.1. A multimodal text-graph dataset of medical codes

The medical codes dataset consists of medical codes, their descriptions, and associated knowledge subgraphs, encompassing eight commonly used health coding systems: ICD-9-CM (procedures and diagnoses), ICD-10-CM, ICD-10-PCS, NDC (National Drug Codes), SNOMED CT, ATC (Anatomical Therapeutic Chemical Classification), CPT (Current Procedural Terminology), and RxNorm. All code lists were obtained from official sources. Specifically, ICD-9 and ICD-10 (CM and PCS) were sourced from the CMS website; NDC codes from the U.S. Food and Drug Administration (FDA) database; and CPT (Level I HCPCS) from the Physician Fee Schedule (PFS) Relative Value Files at CMS. SNOMED CT, RxNorm (active codes only), and ATC were downloaded via the National Library of Medicine (NLM), part of the National Institutes of Health (NIH).

### A.1.1. BUILDING A KNOWLEDGE GRAPH OF MEDICAL CODES

In the final dataset, each medical code is linked to a knowledge graph capturing relevant medical insights and relationships. We constructed these subgraphs in two steps: mapping each code to one or more nodes in the PrimeKG knowledge graph (Chandak et al., 2023); and extracting node-centered subgraphs to represent the code's associated knowledge and connections. To facilitate mapping, we leveraged several external resources, notably the UMLS database (Bodenreider, 2004) and MONDO Disease Ontology files (Balsa-Canto et al., 2023). Medical codes were first mapped to Concept Unique Identifiers (CUIs) in the UMLS database, then linked to PrimeKG nodes via a custom UMLS-to-PrimeKG file. Because PrimeKG includes MONDO annotations, we also aligned medical codes to MONDO terms using the mondo.owl file, thus achieving direct integration with PrimeKG nodes. Additionally, a custom entity linker was employed to enhance coverage by translating medical codes into descriptive text (via PyHealth's MedCode InnerMap) and matching these descriptions to PrimeKG node names. When exact matches were unavailable, we resorted to an NLP-based linker (SciSpacy with UMLS) to measure semantic similarity. For drug codes, the rxnav.nlm.nih.gov API was used to map RxNorm codes to ATC identifiers, which were then associated with DrugBank entities (Wishart et al., 2018) through a predefined ATC-to-DrugBank mapping.

### A.1.2. ASSEMBLING TEXT DEFINITIONS OF MEDICAL CODES

Initially, each medical code's description was taken from its official source. For medication codes (e.g., NDC) where the original text was sparse, additional details were derived from attributes such as trade name, proprietary name, and pharmacological classification. These preliminary definitions were then refined and enriched using GPT-4 (turbo), with prompts tailored to each coding system but sharing a common goal of elaborating on clinical uses (for drugs), procedural steps (for procedures), or mechanistic and clinical context (for diagnoses).

# B. Implementation details

## B.1. Experimental environments

MEDTOK is training on a machine equipped with 4 NVIDIA H100. All experiments were conducted with 1 NVIDIA H100.

We implement MEDTOK using Python 3.9.19, PyTorch 2.3.1, Transformers 4.43.1, and Tokenizers 0.19.1. All LMs and LLMs adopted in this study are retrieved from Hugging Face, except for OpenAI models.

## B.2. Details in MEDTOK training

MEDTOK is trained on 4 NVIDIA H100 GPUs by using the loss defined in Section 3.2. During the training stage, we set the training step as 3000 with a global batch size of 1024, the dimension of quantized vectors is 64. In terms of the models' weights, we freeze the text encoder in MEDTOK and the graph encoder is trainable during the training stage.

## B.3. Implementation details of baseline models

All results presented in this study were obtained using the same machine on which the MEDTOK was trained.

ETHOS (Renc et al., 2024) experiments were conducted using the authors' original repository. For each experimental setting, three models were trained on the MIMIC-IV dataset with different random seeds, and their predictions were averaged during inference to ensure robustness. In the "MEDTOK + ETHOS" configuration, the original vocabulary was extended to

incorporate MEDTOK's tokens for diagnoses, procedures, and prescriptions. The lab measurements were excluded from the analysis. Training and dataset splitting on MIMIC-IV adhered to the methodology outlined in the ETHOS paper. During inference, the number of generated tokens was limited to 2048, and the timeline duration was adjusted based on the specific task: fifteen days for readmission, two weeks for mortality, and up to six months for other tasks. Each model was executed five times, and the resulting predictions were averaged to produce a continuous output, as described in the ETHOS study. Inference on the MIMIC-III dataset was performed on the entire dataset, excluding BMI, ICU stay tables, blood pressure, and lab data. For the EHRShot dataset, inference was conducted on the full dataset for mortality and disease-related tasks, and on randomly selected, stratified samples of ten thousand instances for other tasks.

As for the other baselines adopted in this work, we first downloaded their code and deploy these models on our working machine. For BEHRT (Li et al., 2020) and GT-BEHRT (Poulain & Beheshti, 2024), we re-trained it in an end-to-end way and integrated the tokens for time, visit, and patient's info as that in their original work. For MulT-EHR (Chan et al., 2024), we first pre-train it on MIMIC-III, MIMIC-IV, and EHRShot, respectively, to get the embedding of medical codes, and next fine-tune it on multi-task learning. For the "MEDTOK+" experiments, we use our token embeddings to initialize the nodes or tokens the original work adopted and then train or pre-train the model. It should be noted that we adopt a unified epoch number for all baselines, which is 50.

## C. Task definitions and data preparation under in-patient setting

### C.1. Mortality prediction

**Task definition.** Mortality (MT) prediction estimates the mortality label of the *subsequent* visit for each sample, with the last sample dropped. Formally,

$$f : (v_1, v_2, \ldots, v_{t-1}) \ \rightarrow \ y[v_t],$$

where $y[v_t] \in \{0, 1\}$ is a binary label indicating the patient's survival status recorded in visit $v_t$.

### C.2. Readmission prediction

**Task definition.** Readmission prediction checks if the patient will be readmitted to the hospital within $\sigma$ days. Formally, $f : (v_1, v_2, \ldots, v_{t-1}) \ \rightarrow \ y[\tau(v_t) - \tau(v_{t-1})]$, where $y \in \{0, 1\}$ and $\tau(v_t)$ denotes the encounter time of visit $v_t$. Specifically,

$$y[\tau(v_t) - \tau(v_{t-1})] \ = \ \begin{cases} 1 & \text{if } \tau(v_t) - \tau(v_{t-1}) \leq \sigma, \\ 0 & \text{otherwise.} \end{cases}$$

In our study, we set $\sigma = 15$ days.

### C.3. Length-of-Stay (LOS) prediction

**Task definition.** Length-of-Stay (LOS) prediction follows the formulation of (Harutyunyan et al., 2019), estimating ICU stay length for each visit. Formally, $f : (v_1, v_2, \ldots, v_t) \rightarrow y[v_t]$, where $y[v_t] \in \mathbb{R}^{1 \times C}$ is a one-hot vector indicating its class among $C$ possible categories. We define 10 classes, $\{0, 1, \ldots, 7, 8, 9\}$, representing the following durations: 0 for one day or less, 1-7 for within one week, 8 for one to two weeks, and 9 for at least two weeks.

### C.4. Phenotype prediction

**Task definition.** Phenotype prediction aims to classify which acute care conditions are present in a given patient record: $f : (v_1, v_2, \ldots, v_t) \rightarrow y[v_t]$, where $y[v_t] \in \mathbb{R}^{1 \times C}$ is a one-hot vector indicating its class among $C$ possible categories. This task is a multilabel classification problem with macro-averaged AUC-ROC being the main metric.

### C.5. Drug recommendation

**Task definition.** Drug recommendation aims to recommend drugs for a patient according to the patient's visit history and diagnosis in current visit: $f : (v_1, v_2, \ldots, v_t) \rightarrow y[v_t]$, where $y[v_t] \in \mathbb{R}^{1 \times C}$ is a one-hot vector indicating its class among $C$ possible categories. This task is a multilabel classification problem with macro-averaged AUC-ROC being the main metric.

**Data preprocessing.** In this study, we adopted a data preprocessing approach similar to that used in previous research (Harutyunyan et al., 2019), which defined 25 acute care conditions. Each diagnosis code was mapped to one of these 25 phenotype

categories. Finally, we got 24 diagnosis codes. Since ICD-9 codes in MIMIC-III are associated with hospital visits rather than specific ICU stays, we linked diagnoses to ICU stays using the hospital admission identifier. To reduce ambiguity, we excluded hospital admissions involving multiple ICU stays, ensuring that each diagnosis corresponded to a single ICU stay per admission. It's important to note that our phenotype classification was retrospective; we analyzed the complete ICU stay before predicting the presence of specific diseases.

### C.6. Out-patient setting

Under this setting, we adopt two types of tasks in EHRShot: operational outcomes prediction and assignment of new diagnoses. In the field of operational outcomes, we follow the same task definitions in long length of stay prediction, which only considers if a patient stay in the hospital for less than 7 days or more than 7 days. In terms ofthe readmission task, we set the time window as 15 days, which is the same as that under in-patient setting. We also add another operational outcome task, which is mortality prediction. The definition of mortality prediction is the same as that under the in-patient setting.

