# OpenReview forum: "Multimodal Medical Code Tokenizer"
_ICML.cc/2025/Conference — ICML 2025 poster_

### Official Review · Reviewer_Eig7 · 2025-03-10

**Overall Recommendation:** 4

**Summary:**

MEDTOK is a tokenizer designed specifically for medical codes, improving upon traditional approaches (that treat each code as isolated textual units) by considering the textual description of each medical code and its ontological hierarchy and relationships across different medical coding standards. It employs a language model encoder to process textual descriptions and a graph encoder to represent relational structures among medical codes. Both modalities—textual and relational—are combined into a unified token space suitable for input into various medical models.

Integrating MEDTOK into state-of-the-art EHR models shows substantial improvements across diverse medical tasks, including outcome prediction, diagnosis classification, drug recommendation, and risk stratification.

**Claims And Evidence:**

The paper clearly identifies shortcomings in standard EHR tokenizers and effectively motivates the need for MedTok. The claims about improved performance by incorporating domain knowledge through code descriptions and relational structures are convincingly supported by empirical results. The experiments showing improvements simply by replacing standard tokenizers with MedTok are particularly compelling.

**Essential References Not Discussed:**

The paper seems to cover well relevant related works.

**Experimental Designs Or Analyses:**

The experimental analyses appear sound and clearly demonstrate improvements over state-of-the-art models. The provided ablation studies are valuable, though an additional ablation study excluding cross-modality embeddings would be beneficial to fully understand their contribution.

**Methods And Evaluation Criteria:**

The methods proposed, including leveraging both language models and graph encoding to integrate textual and relational data, are well justified for the medical domain. The evaluation criteria using standard benchmarks like MIMIC-III, MIMIC-IV, and EHRShot are appropriate and relevant.

**Other Comments Or Suggestions:**

Typo:

Line 303-right column: "HRShot" -> "EHRShot"

**Other Strengths And Weaknesses:**

Strengths:

* Clearly motivated and well-presented rationale for MedTok.

* Straightforward integration with existing state-of-the-art models.

* Robust empirical support showing improvements across various medical prediction tasks.

Weaknesses:

* Graph construction methodology seems complex and potentially difficult to reproduce.

* It would enhance the paper to include visualizations of the learned embeddings to better illustrate how MedTok captures relational and textual information.

**Questions For Authors:**

1. How did you assess the quality of the final constructed knowledge graph? Were quantitative or qualitative evaluations performed?

2. Did you conduct ablation studies specifically targeting each step detailed in appendix A.1.1 regarding graph construction? Are all these steps critical?

3. Is the code for graph construction and MedTok tokenizer publicly available or planned to be released?

4. Have you done an ablation study excluding the cross-modality embeddings?

**Relation To Broader Scientific Literature:**

The paper’s contributions clearly extend existing literature on EHR tokenizers by introducing multimodal embeddings leveraging both textual descriptions and relational contexts. This approach offers a meaningful advancement compared to previous studies focusing primarily on isolated textual tokens.

**Theoretical Claims:**

This paper does not explicitly present theoretical proofs.

---

> ### Author Rebuttal · Authors · 2025-04-01
>
> ### **W1:**
> We apologize for the misunderstanding. In the final version, we will clarify that our graph construction approach is relatively simple and straightforward. Specifically, for each medical code, we define the code’s graph as a subgraph of a knowledge graph centered on the node representing the code plus its 2-hop neighborhood. Therefore, graph construction is straightforward and computationally efficient.
>
> ### **W2:**
> Thank you for your insightful feedback. The figure (https://anonymous.4open.science/r/MedTok-8DEE/drug_embedding.png) illustrates different drug codes (NDC, ATC, RxNorm) for the five selected drugs, demonstrating how MedTok effectively captures drug semantics. We observe that different codes representing the same drug cluster together, indicating that MedTok learns meaningful tokens of drugs across various coding systems.
>
> Additionally, NDC and RxNorm are highly granular, meaning a single drug can have multiple codes reflecting different dosages or formulations. Despite this granularity, MedTok successfully groups related tokens, preserving their underlying medical meaning. This highlights MedTok’s ability to generalize across different drug coding schemes.
>
> ### **Comments:**
> Thank you for your careful review. We will change it to ‘EHRShot’ in the next version.
>
> ### **Q1:**
> Thank you for your comment. We ensure the accuracy of the knowledge by extracting subgraphs from a well-established biomedical knowledge graph (e.g., PrimeKG). We include nodes within two hops of the medical code to build the subgraph. However, since PrimeKG is dense, we use the PageRank algorithm to select the top 2,000 most influential nodes connected to the central medical code. Based on these processes, we could make sure the extracted subgraph is of high quality.
>
> ### **Q2:**
> Thank you for your comment. Appendix 1.1 explains how we extract the subgraph centered on the medical code from PrimeKG, including using CUI code to map medical code and entities in KGs. It is about extracting knowledge from an existing KG, not constructing a new KG.
>
> ### **Q3:**
> Yes\! We will release our tokenizer and all source data, including code description and corresponding subgraph.
>
> ### **Q4:**
> Thank you for your valuable suggestion. **Please refer to the responses to Reviewer EVKN: E2-E3 .**
>
> ---
> *We sincerely thank the reviewer for the thoughtful and encouraging feedback on the originality, quality, clarity, and significance of MedTok. We appreciate your positive evaluation of our work and the opportunity to address your comments. In response, we have provided detailed explanations to address your questions. Please do not hesitate to let us know if additional clarifications would be helpful. Thank you again for your valuable input.*

---

> > ### Comment · Reviewer_Eig7 · 2025-04-03
> >
> > Thanks for your replies and additional figure/experiments. I will keep my accept score

---

> > > ### Author Response · Authors · 2025-04-03
> > >
> > > Dear Reviewer Eig7,
> > >
> > > Thank you for your kind words on MedTok. We sincerely appreciate your insightful suggestions on MedTok. Thank you again for your support and for recognizing the value of our research.
> > >
> > > Best,
> > > Authors

---

### Official Review · Reviewer_P3y8 · 2025-03-12

**Overall Recommendation:** 3

**Summary:**

Medical codes in electronic health records (EHRs) contain rich textual descriptions and structured relationships. However, existing tokenization methods treat them as isolated textual tokens, failing to capture their ontological and relational context. The authors propose MEDTOK, a multimodal medical code tokenizer that creates a codebook to select the top k modality-specific or cross-modality embedding tokens, integrating both text-based and graph-based representations of medical codes. They evaluate MEDTOK across five clinical tasks on MIMIC-III and MIMIC-IV, as well as two additional tasks on EHRShot, and conduct an ablation study to demonstrate the effectiveness of the proposed multimodal approach.

**Claims And Evidence:**

- The claims presented in the paper are supported by experimental results and theoretical proofs.
 - Please also check the __Theoretical Claims:__

**Essential References Not Discussed:**

Are there related works that are essential to understanding the (context for) key contributions of the paper, but are not currently cited/discussed in the paper? Be specific in terms of prior related findings/results/ideas/etc. (Example: "The key contribution is a linear-time algorithm, and only cites a quadratic-time algorithm for the same problem as prior work, but there was also an O(n log n) time algorithm for this problem discovered last year, namely Algorithm 3 from [ABC'24] published in ICML 2024.")

   - Line 112 left column "Directly using the tokenizers for languages risks flattening the relationships among codes and failing to preserve the biomedical information", please add reference for this sentence.
   - Line 122 left column "However, graph tokenizers may suffer from information loss when applied to graphs in other domains, please add reference

**Experimental Designs Or Analyses:**

- I have reviewed the experimental designs carefully. The authors thoroughly evaluate their MEDTOK approach using five tasks across two inpatient datasets and two additional tasks on the EHRShot dataset. They also comprehensively compare their proposed tokenizer with alternative tokenizers, such as the VQGraph tokenizer and a standard BERT-based tokenizer.
- However, I have some concerns regarding certain experimental analyses:
 - - In the ablation experiments, MEDTOK demonstrates strong performance. However, simply removing the text-specific embeddings $e_t^s$ or graph-specific embeddings $e_g^s$ might be insufficient to fully assess the impact of each modality, as the model still retains shared embeddings $e_t^c$ and $e_g^c$. Have you tested an ablation setting where only $e^c$ embeddings** are used (i.e., removing both $e_t^s$ and $e_g^s$)? This would provide further insight into how much information is encoded in the cross-modality tokens alone.
 - - Additionally, in the codebook size analysis, there is an unexpected performance drop on both MIMIC datasets when the codebook size is around 18,000. Could the authors provide an explanation or hypothesis for this anomaly? Specifically, this sudden drop is quite substantial, with the AUPRC dropping to nearly one-third compared to when the codebook size is around 6,000. Clarifying this would strengthen the analysis.

**Methods And Evaluation Criteria:**

Yes. The authors use two simple linear projection layers for modality-specific embeddings and a cross-attention module for cross-modality embeddings, both of which are common and appropriate choices in multimodal tasks.

**Other Comments Or Suggestions:**

N/A

**Other Strengths And Weaknesses:**

__Strengths:__
 - The paper is clearly written, making it easy to follow and understand.
 - The authors validate their approach across seven tasks using three datasets. The experimental results outperform all baseline methods. The work is solid.

__Weakness:__ please check the __Experimental Designs Or Analyses__ and __Theoretical Claims__.

**Questions For Authors:**

- Could the authors clarify how the modality-specific embeddings for knowledge graph $e_g^s$ are constructed? While the paper briefly mentions using a linear projector, upon reviewing the provided code, it appears that the graph data might be converted directly into text and then embedded together with other medical textual descriptions.

- Given that this is a multimodal task, could the authors provide a case study or example illustrating situations where one modality compensates for limitations in another?

- Could the authors provide more specific details about how their proposed tokenizer is applied in downstream tasks? While the appendix provides a detailed description of selecting the label $y$, I am interested in how the tokenizer integrates with the inputs in these experiments.

- other questions pleas check the __Experimental Designs Or Analyses__ and __Theoretical Claims__

**Relation To Broader Scientific Literature:**

- The proposed approach extends traditional vector quantization methods by explicitly dividing the codebook into modality-specific and shared regions, which helps preserve modality distinctions while improving cross-modal interactions.

**Theoretical Claims:**

- Yes, I have carefully checked all equations presented in the "Approach" section.
   - The Equations (8) and (9) define modality-specific quantized vectors $\hat{e}_t^s, \hat{e}_g^s$, but they do not appear to be explicitly used in any other equation. Could you please clarify how they contribute to the overall optimization process?
   - In the token packing section, why do the authors use
$\mathcal{L}_{\text{InfoNCE}} (\hat{e}_t^c, \hat{e}_t^c) + \lambda \mathcal{L}_{\text{orthogonal}}(\hat{e}_t^c, e_t^c) + \mathcal{L}_{\text{InfoNCE}}(\hat{e}_g^c, \hat{e}_g^c) + \lambda \mathcal{L}_{\text{orthogonal}}(\hat{e}_g^c, e_g^c)$
to represent $\mathcal{L}_{\text{token}}^{s}$ instead of directly using $\hat{e}_t^s, e_t^s$? Especially considering that the loss function already includes $\mathcal{L}_{\text{token}}^{c}$. While the definition of $\mathcal{L}_{\text{token}}^{c}$ is clear based on Wang et al.'s previous work, the reasoning for the specific formulation of $\mathcal{L}_{\text{token}}^{s}$ is less evident. Could the authors elaborate on the motivation behind choosing this particular formulation?

---

> ### Author Rebuttal · Authors · 2025-04-01
>
> ### **Theoretical Claims:**
> Thank you for your comment. Equations 8-9 contribute to the overall optimization process by representing the optimal values of graph embeddings (Eq. 8\) and text embeddings (Eq. 9), which are designed to model modality-specific and modality-shared information. To approximate these optimal values, we use a method inspired by Wang et al., 2024 ([https://openreview.net/forum?id=r7Xnetd0Pq\#discussion](https://openreview.net/forum?id=r7Xnetd0Pq#discussion)), where the optimization is achieved through InfoNCE loss and orthogonal loss.
>
> The contribution of modality-specific and modality-shared information is shown in the response to **Reviewer EVKN: E2-E3.** By optimizing both shared and specific information across two modalities, the performances are improved by 3.9%, 5.7%, and 9.1% on three datasets, respectively.
>
> ### **E1:**
> **Please refer to the responses to Reviewer EVKN: E2-E3.** By optimizing both shared and specific information across two modalities, the performances are improved by 3.9%, 5.7%, and 9.1% on three datasets, respectively.
>
> ### **E2:**
> One possible explanation for the codebook usage is as follows: at 18,000 tokens, the usage was around 32%, compared to \~40% when the size was 6,000. As the codebook expands, the model may struggle to allocate representation capacity effectively, leading to less efficient learning and potential overfitting to specific token patterns. Such observations could also be observed by other tokenizer work (https://arxiv.org/abs/2308.02117, https://arxiv.org/html/2406.11837v1), which makes this a good open problem and future direction for the field.
>
> ### **References:**
> We add the following two references to strengthen our argument and make our explanation more convincing.
>
> *\[1\] Shang, Junyuan, et al. "Pre-training of graph augmented transformers for medication recommendation." IJCAI. 2019\.*
> *\[2\] Xia L, Kao B, Huang C. Opengraph: Towards open graph foundation models\[J\]. arXiv preprint arXiv:2403.01121, 2024\.*
> ### **Q1:**
> For modality-specific embeddings in the graph, we first use a graph encoder to learn the graph embedding, followed by a projector to learn the graph-specific embedding. The graph-specific embedding is optimized separately using a loss function and is not combined with the medical textual descriptions. Please refer to the 'specific embedding' function (Line 188-218) in [https://anonymous.4open.science/r/MedTok-8DEE/vector\_quantization\_soft\_one\_new.py](https://anonymous.4open.science/r/MedTok-8DEE/vector_quantization_soft_one_new.py).
>
> ### **Q2:**
> Example 1: Heart Failure (I50.9)
> The text description for this code provides information about heart failure, its core diagnostic criteria, and primary clinical features (for example, impaired ventricular function). However, it does not include information on risk factors, complications, and treatment options–critical information that is not included in the text description but exists in the knowledge graph. Complementing this text definition with relational, graph-based information reveals related conditions such as hypertension, coronary artery disease, and diabetes; suggests treatments like beta-blockers, ACE inhibitors, and diuretics; and highlights potential complications, including pulmonary edema and kidney dysfunction.
>
> Example 2: Type 2 Diabetes Mellitus, Without Complications (E11.9)
> The text description captures a type 2 diabetes diagnosis along with lab values (fasting blood glucose and an HbA1c) and notes that the patient is prescribed metformin. However, it does not provide information on long-term risks or alternative treatment options. In contrast, the knowledge graph enriches this snapshot by linking diabetes to related conditions such as obesity, hypertension, and dyslipidemia; by highlighting potential complications like diabetic retinopathy, neuropathy, and chronic kidney disease; and by suggesting additional treatments such as insulin therapy or GLP-1 receptor agonists.
>
> In both examples, the text modality provides an immediate clinical snapshot, while the knowledge graph adds a broader context that supports comprehensive and personalized care decisions.
>
> ### **Q3:**
> The output of MedTok includes both tokens and their corresponding embeddings. Specifically, the token refers to the indices in the codebook, and the embedding is the corresponding vector for that index. When integrating with other models, we first obtain the tokens from MedTok and then use these tokens to query the codebook and retrieve the corresponding embeddings, which are used as input to the other models.
>
> ---
>
> *Thank you for your helpful feedback. If you feel our responses are insufficient to motivate increasing your score, we would love to hear from you further about how we can better address your questions. Thank you again\!*

---

### Official Review · Reviewer_1aYX · 2025-03-13

**Overall Recommendation:** 3

**Summary:**

This paper introduces MEDTOK, a multimodal medical code tokenizer that combines text descriptions and relational information from medical ontologies to improve the processing of medical codes in electronic health records (EHRs). MEDTOK encodes both text and graph information into a unified token space, enhancing model performance on various clinical tasks, such as disease prediction and drug recommendation. Experimental results show MEDTOK’s efficiency across multiple datasets.

**Claims And Evidence:**

The paper shows a strong understanding of the field, with convincing motivation and model design. It highlights the limitations of traditional tokenizers and presents MEDTOK as a solution that combines text and graph encoding, supported by experimental results. However, the proposed loss functions lack ablation experimental validation to prove their effectiveness.

**Essential References Not Discussed:**

The research covers most of the essential related works, especially in the context of tokenization for medical codes in EHRs. It discusses relevant studies on traditional tokenization methods, transformer-based models, and multimodal approaches, providing a solid foundation for its contributions.

**Experimental Designs Or Analyses:**

To demonstrate the effectiveness of MEDTOK, the authors have selected appropriate downstream tasks and datasets that are professionally relevant. The ablation design effectively shows the impact of different modality data and the codebook size on the model's performance. Additionally, the MedicalQA experiment adds depth to the overall experimental design. However, in the generative tasks, the choice of the LLM and the datasets used are relatively limited, authors should expand more datasets to further strengthen the findings and provide a more robust evaluation of MEDTOK’s capabilities in diverse scenarios.

**Methods And Evaluation Criteria:**

The proposed method is novel, combining text and graph encoding to address the limitations of traditional medical code tokenizers. The datasets and downstream tasks selections are well-aligned with domain standards, ensuring the method’s relevance and applicability to real-world clinical scenarios. Tasks designs are critical for evaluating the model’s performance and its potential impact in healthcare applications

**Other Comments Or Suggestions:**

n/a

**Other Strengths And Weaknesses:**

Strengths:
1.	The authors address a practical problem in healthcare, specifically the tokenization of medical codes in EHRs, which has significant real-world relevance.
2.	The proposed MEDTOK shows improvements across various downstream tasks and datasets, demonstrating its potential in real-world applications.

Weaknesses:
1.	Graph-specific, Graph-shared, Text-shared, and Text-specific mentioned in the method lack sufficient explanation, particularly regarding why these features are chosen and how they positively impact the results. This section left me confused. I want the authors include a visualization of these embedding methods and provide a clearer explanation to help readers understand how these different embedding features improve the model's performance. Additionally, authors should supplement the experiments by analyzing the specific impact of different embedding methods (such as Graph-specific and Text-specific) on task results and provide both quantitative and qualitative analyses.
2.	While the paper performs well on the evaluated datasets, authors should explore how MEDTOK scales with even larger or more diverse EHR datasets.
3.	Although the paper compares MEDTOK to standard tokenization methods, it should compare the result with other baseline models, including a comparison with a model where patient electronic health records are directly added to the vocabulary.
4.	The paper mentions an ablation study evaluating the impact of removing the text or graph modality on model performance, but the experimental details and analysis are brief. It lacks a more detailed evaluation of different module combinations and loss functions.

5.	Even though transforming multi-modal information into shared features and unique features has already been applied in many models (e.g., [1] and [2]), the use of graph features and text features from biomedical knowledge graphs in this paper is sufficiently novel, with strong motivation that addresses a critical pain point in medical multimodality.
[1] Multi-modal Learning with Missing Modality via Shared-Specific Feature Modelling
[2] DeCUR: DECOUPLING COMMON & UNIQUE REPRESENTATIONS FOR MULTIMODAL SELF-SUPERVISION

**Questions For Authors:**

Please see the Weakness session

**Relation To Broader Scientific Literature:**

The authors clearly position their work within the context of existing literature, highlighting the challenges faced by traditional methods in tokenizing medical codes in the patient electronic health records (EHRs) context.

**Theoretical Claims:**

The theoretical underpinnings of the proposed modules are presented clearly, with well-defined mathematical formulas that explain the process in detail. The authors effectively justify the design choices, such as the use of modality-specific and cross-modality embeddings, using formal representations to support their approach.

---

> ### Author Rebuttal · Authors · 2025-04-01
>
> ### **Claims**:
> **Please refer to the results in ‘Reviewer EVKN: E2’.** The obtained results demonstrate that both shared and specific information optimization enhance performance, with the full optimization achieving the best results across all datasets.
> ### **E1**:
> To address your concerns, we added two new LLMs to the paper (Qwen2.5-7B and MMedLM), generated tokens using MedTok, and prefix-tuned them on the MedMCQA dataset. We then evaluated these fine-tuned models on three other established QA datasets. Results are:
> ||MMLU|MedDDx|AfrimedQA|
> |:----|:----|:----|:----|
> |Llama3.1-8B|0.634|0.424|0.502|
> |+ MedTok|0.664|0.494|0.540|
> |Qwen2.5-7B|0.667|0.334|0.618|
> |+ MedTok|0.770|0.437|0.657|
> |MMedLM|0.564|0.308|0.491|
> |+ MedTok|0.640|0.403|0.653|
> ### **W1:**
> Graph/text-shared information refers to the shared information between graph and text modalities, while graph/text-specific information refers to information specific to each modality. These features are learned by maximizing the shared information and minimizing the overlap between shared information. By considering both modality-shared and modality-specific information, tokens in MedTok can distinguish medical codes that have similar names (i.e., same text descriptions) but different relational structures in corresponding medical code taxonomies (i.e., different graph structures). For example, consider E11.9 (Type 2 Diabetes Mellitus Without Complications) and E11.12 (Type 2 Diabetes Mellitus With Chronic Kidney Disease). These codes both refer to Type 2 Diabetes but are used for patients with different disease progression. E11.9 is linked to general diabetes management and metabolic pathways, while E11.12 is associated with kidney-specific pathways and nephroprotective drugs. MedTok’s optimization strategy can use the shared and specific information between these codes, which improves modeling of these seemingly similar yet distinct codes with different clinical relevance.
>
> To further address your concerns, we performed new experiments to quantify the impact of modality-shared and modality-specific information on MedTok’s. For that, **please refer to new results in ‘Reviewer EVKN: E2’.**
>
> ### **W2:**
> We evaluated MedTok using three EHR datasets that vary considerably in scale and scope.
> * EHRShot is a longitudinal dataset containing **41.6M observations** from **921K visits** across **6,739 patients** (1909–2023). Patients have an average timeline of **59 years**(max **88**) and **136 visits** (max **2,397**).
> * MIMIC IV includes **185.8M observations**, **546K visits**, and **364K patients**. Patients have **1.5 years** of data on average (max **14.7**) with **2.4 visits** (max **238**).
> * MIMIC III contains **32.9M observations**, **58,976 visits**, and **46,520 patients**. The average record spans **4 months** (max **11.5 years**) with **1.3 visits** (max **42**).
>
> Our results across these three datasets demonstrate strong performance, efficiency, and robustness of MedTok across a wide variety of clinical environments (outcome and diagnostic prediction tasks), both in-patient and out-patient contexts, both acute and chronic medical conditions, and patients with varying volumes of clinical data (small vs. large EHR record).
> ### **W3**:
> That is an excellent suggestion. All five baselines we adopted are models that directly incorporate patient electronic health records into their vocabulary. We compared the performance of these baselines with and without our MedTok tokenizer. Results show that integrating MedTok improves the performance of these baseline models by 3.29%, 2.67%, and 5.01% across three datasets (w2\) relative to using baseline models with standard tokenization methods.
> ### **W4:**
> We have added ablation studies on different module combinations and loss functions to examine the utility of each module and loss function component.  **please refer to the response to Reviewer EVKN: E2-E3.** Results show that the performance of full MedTok increases by 3.9%, 5.7%, and 9.1% across three datasets relative to the simplified version of MedTok with modality-shared and modality-specific modules turned off.
> ### **W5**:
> We appreciate your acknowledging the importance of considering both graph and text features for developing a comprehensive tokenizer of medical codes. As we illustrate in our response to W1 and as you nicely point out, multimodality is particularly relevant to model codes that are similar across both graph-text modalities as well as codes that are similar in one modality but not in the other.
>
> —-
> *Thank you again for your thoughtful commentary. If you feel our responses are insufficient to motivate increasing your score, we would love to hear from you further about how we can better address your concerns. Thank you again\!*

---

### Official Review · Reviewer_EVKN · 2025-03-14

**Overall Recommendation:** 3

**Summary:**

In this paper, the authors present MEDTOK, a multimodal medical code tokenizer that integrates textual descriptions of medical codes with graph-based relational contextual information. The proposal employs separate encoders to process each modality, and the resulting representations are mapped into a shared space through vector quantization, ensuring the preservation of both modality-specific and modality-shared information. The authors evaluate MEDTOK on three electronic health record (EHR) datasets using five different backbone models to assess its effectiveness.

## update after rebuttal
I appreciate the authors' responses to my concerns, particularly the newly supplemented results. Accordingly, I have updated my original assessment.

**Claims And Evidence:**

The proposed MEDTOK functions as a broadly defined "tokenizer" rather than a conventional one. Specifically, its tokenization process integrates both tokenization and embedding through dedicated encoders. Given this design, the extensive body of research on medical code embeddings should be discussed in the related work section and incorporated as baselines in the experimental evaluation to provide a more comprehensive comparison.

Furthermore, MEDTOK primarily adapts and integrates existing techniques, including the designed encoders, vector quantization, and the modality fusion mechanism inspired by Wang et al. (2024a). As a result, the degree of novelty in the proposal is limited.

Additionally, prior studies have extensively leveraged graph-based knowledge information to enhance EHR data analytics, including [a, b, c], among others. The authors should explicitly articulate how MEDTOK differentiates itself from these works in terms of methodology, contributions, and empirical performance. A thorough comparative analysis, both conceptually and through experimental validation, would strengthen the claims of the paper.

[a] Choi, Edward, et al. "GRAM: graph-based attention model for healthcare representation learning." Proceedings of the 23rd ACM SIGKDD international conference on knowledge discovery and data mining. 2017.

[b] Shang, Junyuan, et al. "Pre-training of graph augmented transformers for medication recommendation." IJCAI. 2019.

[c] Burger, Manuel, Gunnar Rätsch, and Rita Kuznetsova. "Multi-modal graph learning over umls knowledge graphs." Machine Learning for Health (ML4H). PMLR, 2023.

**Essential References Not Discussed:**

The paper should discuss related works on medical code embedding and graph-based knowledge exploitation in EHR data (such as [a, b, c]).

[a] Choi, Edward, et al. "GRAM: graph-based attention model for healthcare representation learning." Proceedings of the 23rd ACM SIGKDD international conference on knowledge discovery and data mining. 2017.

[b] Shang, Junyuan, et al. "Pre-training of graph augmented transformers for medication recommendation." IJCAI. 2019.

[c] Burger, Manuel, Gunnar Rätsch, and Rita Kuznetsova. "Multi-modal graph learning over umls knowledge graphs." Machine Learning for Health (ML4H). PMLR, 2023.

**Experimental Designs Or Analyses:**

The evaluation should include existing methods for both medical code embedding and graph-based knowledge exploitation in EHR data as baselines to ensure a more comprehensive comparison.

Several aspects of the experimental evaluation require further discussion or additional experiments to strengthen the findings:

(i) In both the comparison between MEDTOK and the baselines, as well as in the ablation study, only the backbone TransformEHR is used for evaluation. Is TransformEHR the best-performing backbone model? The rationale behind this choice should be explicitly justified.

(ii) Additional ablation studies should be conducted to further analyze the contribution of individual components within MEDTOK.

(iii) The influence of critical hyperparameters, such as $\beta$ and $\lambda$ in the loss functions, should be investigated to assess their impact on the performance of MEDTOK.

(iv) Given the focus on EHR data analytics, it would be valuable to provide interpretable findings and medical validation to demonstrate how the proposed MEDTOK can benefit healthcare practitioners in real-world applications.

**Methods And Evaluation Criteria:**

The selection of datasets for evaluation, specifically MIMIC-III, MIMIC-IV, and EHRShot, is well-justified given their relevance to the problem under investigation.

**Other Comments Or Suggestions:**

In Section 4.2, “HRShot.”

**Other Strengths And Weaknesses:**

**Other Strengths:**
The experimental evaluation encompasses a comparison with two baseline methods across three EHR datasets using five different backbone models, an ablation study on modalities, a hyperparameter sensitivity analysis, and a case study on medical Q&A.

**Other Weaknesses:**
The clarity of the paper could be improved by addressing the following points:

(i) The evaluation setup does not display the 24 phenotypes used for in-patient evaluation.

(ii) In the drug recommendation task for in-patient evaluation, the justification for restricting the recommendation scope to five specific drug candidates should be clearly articulated.

(iii) In Figure 3, for the results on the EHRShot dataset, it is unclear which two tasks out of the seven are selected for demonstration.

(iv) In Appendix B.3, the rationale for conducting inference on sampled datasets rather than the entire test set for certain tasks on the EHRShot dataset should be explicitly explained.

**Questions For Authors:**

Please refer to my detailed comments and suggestions outlined in the questions above.

**Relation To Broader Scientific Literature:**

The proposed MEDTOK builds on prior research in multimodal learning for EHR analytics by leveraging the intra-modality and inter-modality relationships to improve predictive performance.

MEDTOK extends these ideas by introducing a multimodal tokenization component for encoding different modalities separately and a token packing mechanism to integrate complementary information. This structured approach enhances analytic performance, contributing to the advancement of EHR-based predictive modeling.

**Theoretical Claims:**

I have reviewed the theoretical claims in detail.

---

> ### Author Rebuttal · Authors · 2025-04-01
>
> ### **Claims:**
> We added the following discussion to related work: *"Rather than treating medical codes in isolation, some methods incorporate additional knowledge to enhance their representation using structures like knowledge graphs (Choi et al., 2017; Burger et al., 2023\) or ontology trees (Shang et al., 2019). These methods build relationships between medical codes, improving performance on EHR tasks."*
>
> MedTok differs from existing methods:
> * MedTok is a **tokenizer**, converting raw input into **discrete tokens** for transformer models. It doesn't directly learn high-dimensional embeddings for downstream tasks but maps **inputs to a fixed set of tokens**.
> * MedTok maintains a codebook during training, unlike traditional representation learning methods that maintain a set of medical code embeddings.
>
> To address concerns, we analyzed alternative methods and integrated MMUGL (Burger et al., 2023\) as MedTok’s encoder, training it with a vector quantization loss. We compared MMUGL-driven MedTok with Transformer-based models on EHR tasks. GRAM (Choi et al., 2017\) was excluded due to its dependence on EHR settings, and G-Bert (Shang et al., 2019\) only considers medication and diagnostic codes, which do not align with our setting. Results are:
> ||MIMIC III|MIMIC IV|EHRShot|
> |:----|:----|:----|:----|
> |MMUGL-MedTok|0.370|0.305|0.361|
> |MedTok|0.412 (+4.2%)|0.444 (+13.9%)|0.378 (+1.7%)|
> ### **E1:**
> MedTok can be used with any transformer-based EHR predictive model. Our benchmarking included comparisons with five such models. TransformEHR is the best-performing backbone across 4 tasks, and thus it was chosen.
> ### **E2:**
> MedTok uses two modalities and results show optimal performance when using both modalities (Fig. 4). We additionally conduct ablation studies, demonstrating that shared and specific information optimization enhances performance:
> ||MIMIC III|MIMIC IV|EHRShot|
> |:----|:----|:----|:----|
> |VQ|0.373|0.387|0.287|
> |VQ \+ shared|0.379|0.409|0.314|
> |VQ \+ specific|0.382|0.402|0.366|
> |VQ \+ shared \+ specific|0.412|0.444|0.378|
> ### **E3:**
> Following your advice, we examine the impact of hyperparameters on MedTok performance. To make MedTok consider shared and specific information equally, we assume λ\= β, where λ is the weight for shared information loss and β for specific information. Results are:
> |λ \= β|MIMIC III|MIMIC IV|EHRShot|
> |:----|:----|:----|:----|
> |0.01|0.356|0.376|0.425|
> |0.1|0.412|0.444|0.378|
> |0.2|0.344|0.388|0.381|
> |0.3|0.330|0.357|0.418|
> |0.4|0.382|0.327|0.409|
> |0.5|0.403|0.401|0.404|
> ### **E4:**
> To this end, we selected a subset of patients classified as high risk for Hyperlipidemia by MedTok \+ TransformEHR, where these patients have no records of Hyperlipidemia before. We then counted the tokens assigned to these patients and identified those appearing more than 100 times (https://anonymous.4open.science/r/MedTok-8DEE/E4.png). We then mapped these frequent tokens to medical codes, with the most overlapping codes being Rosuvastatin 5 mg Oral Tablet (RxNorm 2669980), Burn of skin (SNOMED CT 147087003), Type 2 diabetes mellitus without complication (disorder) (SNOMED CT 373555004), and Hyperlipidemia (SNOMED CT 285605009). They are closely related to Hyperlipidemia. Rosuvastatin corresponds to medications commonly prescribed for lipid disorders. The other three medical codes represent clinical diagnoses or findings associated with hyperlipidemia-related cardiovascular risk. It suggests that MedTok effectively captures key medical concepts related to Hyperlipidemia, supporting its predictive capability.
> ### **W1:**
> We identify 24 phenotypes following (Harutyunyan et al., 2019). Please refer to Table 2 in https://doi.org/10.1038/s41597-019-0103-9 for details.
> ### **W2:**
> We limited the scope to five drugs to assess MedTok on well-defined, clinically relevant drug recommendations across diverse therapies. Each selected drug was prescribed to ~20–30% of patients, underscoring their relevance in clinical decision-making. Besides, the selection spans a broad range of categories: antibiotics (Vancomycin and Levofloxacin), anticoagulant (Heparin Sodium), beta-blocker (Metoprolol), and lipid-lowering agent (Atorvastatin). This range ensures that the evaluation covers multiple clinical scenarios, while focusing on conditions readily identifiable from patient data.
> ### **W3:**
> We follow https://arxiv.org/abs/2307.02028 in categorizing seven tasks. Operational Outcomes (OO) includes length of stay, readmission, and mortality prediction, while Assignment of New Diagnoses (ND) includes four new disease diagnoses.
> ### **W4**
> Since EHRShot is a longitudinal dataset with 921,499 visits and per-visit readmission predictions, we use stratified sampling to reduce redundancy while ensuring consistency with the ETHOS training setting.
> ### **Comments:**
> We revise it to ‘EHRShot’.
>
> ---
> We appreciate your suggestions for improvement. Please reach out with any questions or for further clarification. Thank you\!

---

> > ### Comment · Reviewer_EVKN · 2025-04-03
> >
> > I would like to thank the authors for their rebuttal, especially for the newly added experimental results. In light of this, I am pleased to raise my rating accordingly.

---

> > > ### Author Response · Authors · 2025-04-03
> > >
> > > Dear Reviewer EVKN,
> > >
> > > We are especially grateful for your recognition of our contributions and for the increased score. We sincerely appreciate your insightful question and valuable suggestions. Those are extremely useful and make MedTok more comprehensive and solid. Thank you again for your support and we deeply appreciate your expertise!
> > >
> > > Authors

---

### Decision · Program_Chairs · 2025-05-01

**Decision:**

Accept (poster)

**Comment:**

The paper proposes a multimodal medical code tokenizer that uses the text descriptions and relational context of codes to tackle patient electronic health records. The motivation is meaningful, and the method is well-estimated. During the review and rebuttal periods, all reviewers' concerns are solved. Therefore, I recommend the acceptance.